# Endolysins as Effective Agents for Decontaminating *S. typhimurium*, *E. coli*, and *L. monocytogenes* on Mung Bean Seeds

**DOI:** 10.3390/ijms26052047

**Published:** 2025-02-26

**Authors:** Fangfang Yao, Jiajun He, Raphael Nyaruaba, Hongping Wei, Yuhong Li

**Affiliations:** 1State Key Laboratory of Oral & Maxillofacial Reconstruction and Regeneration, Key Laboratory of Oral Biomedicine Ministry of Education, Hubei Key Laboratory of Stomatology, School & Hospital of Stomatology, Wuhan University, Wuhan 430079, China; 2CAS-Key Laboratory of Synthetic Biology, CAS Center for Excellence in Molecular Plant Sciences, Institute of Plant Physiology and Ecology, Chinese Academy of Sciences, Shanghai 200032, China; 3University of Chinese Academy of Sciences, Beijing 100039, China; 4Key Laboratory of Virology and Biosafety, Wuhan Institute of Virology, Chinese Academy of Sciences, Wuhan 430071, China

**Keywords:** food safety, sprout, mung bean contamination model, endolysin, *Salmonella*, *Escherichia coli*, *Listeria monocytogenes*

## Abstract

Seeds are a major source of contamination by foodborne pathogens such as *Salmonella typhimurium*, *Escherichia coli*, and *Listeria monocytogenes*, significantly increasing the risk of foodborne diseases associated with fresh produce like sprouts. In this study, we described novel endolysins and the engineered variants that exhibited potent bactericidal activity against these pathogens. These endolysins demonstrated strong bactericidal effects independently of outer membrane permeabilizers, effectively killing *S. typhimurium*, *E. coli*, and *L. monocytogenes* to undetectable levels (>4-log kill) at concentrations as low as 12.5 μg/mL. The enzymes retained their activity in complex environments, such as a wide range of temperatures (4–100 °C), pH values (4–10), serum concentrations (0–50%), and sodium chloride concentrations (0–500 mM). Furthermore, their rapid bactericidal kinetics, excellent storage stability (>18 months), and broad-spectrum antimicrobial activity enhanced their potential for application. These endolysins remained effective against stationary-phase bacteria and biofilm-forming bacteria, achieving more than 99% biofilm eradication at 200 μg/mL. Notably, at concentrations as low as 50 μg/mL, these enzymes completely decontaminated foodborne pathogens in a mung bean seed model contaminated with 4–5 log CFU of bacteria. This study is the first to report the successful use of lysins to control both Gram-negative and Gram-positive pathogens on mung bean seeds.

## 1. Introduction

Foodborne illness is among the most pressing public health concerns worldwide. Each year, unsafe food causes 600 million cases of foodborne illnesses and 420,000 fatalities worldwide, with 30% of these deaths occurring in children under the age of five [1]. In the United States, multistate foodborne disease outbreaks caused by fruits and vegetables have steadily increased since 1973, gradually surpassing those associated with meat [2]. According to a survey of bacterial foodborne illness outbreaks associated with fresh fruits and vegetables in the United States, sprouts are identified as the most common type of vegetable involved [3]. Sprouts are a traditional food with a long history and have gained increasing popularity among consumers due to their freshness, sustainable production, and richness in bioactive compounds [4,5,6]. However, due to the consumption of sprouts which are often in their raw form, there is a lack of a high-temperature cooking process to eliminate bacteria, which increases the risk of foodborne illnesses [3,7,8]. The most common pathogens associated with these outbreaks are *Salmonella* spp. and *Escherichia coli* [9]. In 2010, an outbreak of a Shiga-toxin-producing *E. coli* contamination in sprouts occurred in Germany, resulting in nearly 4000 cases, and hemolytic uremic syndrome developed in more than 20% of the identified cases [10]. Although there have been only a few outbreaks associated with *Listeria*-contaminated sprouts, *Listeria* warrants special attention. Among the 31 major foodborne pathogens, *Listeria* had the highest hospitalization rate (94%) and a high mortality rate (15.9%) [11]. A 2015 outbreak in the United States associated with a *Listeria* contamination of mung bean sprouts involved five individuals across two states, Illinois and Michigan, all of whom were hospitalized, with two fatalities [12].

Most sprout-related foodborne illness outbreaks can be traced back to seed contamination [11]. Pathogen contamination in seeds may occur during the preharvest stage due to agricultural soil, irrigation water, or manure contamination [13]. Additionally, human and mechanical contact during the harvesting process can introduce new contaminants, which may persist or even proliferate during storage [9]. The sprout production process involves germinating seeds in a warm, moist environment for 3–6 days, which allows for even a small number of pathogens present in the seeds to proliferate rapidly to pathogenic levels [14]. Thus, seed disinfection is considered a critical step in microbial control during sprout production [8,15]. The Canadian Food Inspection Agency (CFIA) recommends disinfecting seeds prior to sprouting to achieve a minimum 3-log reduction in the number of microbial pathogens of concern [16]. Various methods have been explored to inactivate pathogens on seeds, including chemical, physical, and biological approaches [8,17]. These methods show promising potential, such as calcium hypochlorite (20,000 ppm) [15], irradiation [18], plasma [19], lactic acid bacteria [20], and bacteriophages [21,22]. However, their application in the food industry faces several limitations, including worker safety concerns (chlorine compounds) [15], high commercial costs (plasma) [23], low efficacy (irradiation) [17,23], strict safety requirements (lactic acid bacteria) [20], and bacterial resistance (bacteriophages) [24,25].

Endolysins are peptidoglycan hydrolases encoded by bacteriophages that can degrade bacterial cell walls, leading to bacterial death [26]. As an emerging and promising natural antimicrobial agent, endolysins, unlike bacteriophages, pose no risk of gene transfer and do not induce bacterial resistance [26,27]. Many highly active Gram-positive bacteria endolysins have been discovered. However, these endolysins typically have a very narrow lytic spectrum, mostly being effective against certain strains of specific species. Nevertheless, Gram-positive bacterial endolysins have been successfully applied in clinical settings and the food industry (milk, cheese, and meat), such as for controlling *Staphylococcus aureus* [28,29,30]. On the other hand, in Gram-negative bacteria, the peptidoglycan is enclosed by a dense lipopolysaccharide outer membrane, which hinders endolysin access to the peptidoglycan, limiting their effectiveness [31]. Many efforts have focused on combining endolysins with various outer membrane permeabilizers, such as EDTA and poly-L-lysine, to facilitate membrane penetration, which limits their potential application [32]. Some studies have utilized antimicrobial peptides (AMPs), fusing them to the N- or C-terminus of endolysins through protein engineering, enabling endolysins to penetrate the outer membrane and enhance bactericidal activity [33,34,35]. In fact, some natural endolysins have demonstrated high intrinsic activity against clinical Gram-negative pathogens, such as *Acinetobacter baumannii* [36] and *Pseudomonas aeruginosa* [37]. Unfortunately, for key foodborne pathogens in Gram-negative bacteria, such as *Salmonella* and *E. coli*, reported endolysins typically exhibit bactericidal activity only after chloroform treatment to permeabilize the outer membrane (e.g., LysSTP4 [38], LysSE24 [39], LysPB32 [40], and LysF1 [41]), or require the use of additional outer membrane permeabilizers to achieve bactericidal activity (e.g., LysSP1 [42], SPN1S [43], Salmcide-p1 [44], XFII [45], and LysECP26 [46]). There is currently a significant shortage of endolysins with high intrinsic bactericidal activity against *Salmonella* and *E. coli*, which limits the application of this promising antimicrobial agent in the food industry.

This study aims to explore the potential application of endolysins in controlling microorganisms during sprout production. We designed and developed novel endolysins and their engineered variants that did not rely on outer membrane permeabilizers, and evaluated their antimicrobial efficacy in vitro against major foodborne pathogens, including *S. typhimurium*, *E. coli*, and *Listeria monocytogenes*. A mung bean model was used to simulate real food environments to assess the biocontrol potential of these endolysins in simultaneously inhibiting *S. typhimurium*, *E. coli*, and *L. monocytogenes*. By investigating innovative strategies for controlling these pathogens in food, this study seeks to provide novel alternatives and enhance food safety.

## 2. Results

### 2.1. Evaluation of Endolysin Activity Against S. typhimurium, E. coli, and L. monocytogenes

The bactericidal activities of novel endolysins discovered in our laboratory were evaluated against *S. typhimurium*, *E. coli*, and *L. monocytogenes* using a log-killing assay. The results showed that all endolysins reduced the viability of *S. typhimurium* CMCC (B) 50115 and *E. coli* O157:H7 to below the detection limit (<100 CFU/mL) (Figure 1a). Additionally, LysPd078 and LysPd138 effectively reduced the viability of *L. monocytogenes* WHG50001 to below the detection limit. In contrast, LysPd144 and LysPd157 reduced the *L. monocytogenes* WHG50001 CFU count by only 1 to 2 log within 1 h (Figure 1a). We further assessed the peptidoglycan degradation activity of these endolysins using a halo assay. Clear zones were observed on the lawns of autoclaved *S. typhimurium* CMCC (B) 50115 and *E. coli* O157:H7 for all endolysins tested (Figure 1b). However, only LysPd078 and LysPd138 produced clearing zones on *L. monocytogenes* WHG50001. Next, we further characterized their biochemical properties using *S. typhimurium*.

### 2.2. Characterization of Endolysins

To evaluate the relative killing rate of the endolysins, we incubated *S. typhimurium* CMCC (B) 50115 cells with these endolysins. The time-killing assay showed that LysPd144 rapidly killed *S. typhimurium*, resulting in a ~3 log reduction within 5 min and a reduction to below the detection level after 30 min (Figure 2a). Further experiments demonstrated that the lytic activities of these endolysins were dose-dependent (Figure 2b). LysPd078 and LysPd144 reduced the viability of *S. typhimurium* to undetectable levels at concentrations as low as 25 μg/mL. We next compared the activity of the endolysins against log-phase and stationary (grown overnight) *S. typhimurium* cells (Figure 2c). It was established that the stationary-phase bacteria were more resilient than the log-phase cells. Nonetheless, LysPd078 retained significant killing activity against stationary-phase *S. typhimurium*.

We further evaluated the bactericidal activity of the endolysins under various conditions. The addition of NaCl inhibited the activity of most Gram-negative endolysins; however, LysPd144 retained some activity even at NaCl concentrations as high as 150 mM (Figure 3a). We also assessed the impact of pH on endolysin activity. Remarkably, LysPd078 demonstrated a superior bactericidal efficacy across a wide pH range (pH 4–10), reducing *S. typhimurium* to below the detection limit (Figure 3b). Although protein-based antimicrobials often exhibit a limited thermal stability, these endolysins maintained high activity across all temperatures tested, including at 100 °C (Figure 3c). Moreover, LysPd078 retained 100%, 99.9%, and over 92.7% of its bactericidal activity after 18 months of storage at 4 °C, 22 °C, and 37 °C, respectively (Figure 4). Based on these promising results, we selected LysPd078 for further experiments.

Next, we evaluated the bactericidal activity of LysPd078 against various *Salmonella*, *E. coli*, and *Listeria* species (Figure 5). After 1 h of incubation, LysPd078 demonstrated an exceptionally broad-spectrum bactericidal effect against all bacterial strains tested. In particular, LysPd078 showed strong bactericidal activity against all *Salmonella* and *E. coli* strains, reducing the colony count of these strains to below the detection limit.

### 2.3. Bactericidal Activity of Engineered LysPd078

Although LysPd078 demonstrated a high bactericidal activity, its reduced lytic efficacy in the presence of salt posed a challenge for its broader application. To enhance its lytic activity, different peptides were fused on the N-terminal of LysPd078, resulting in engineered variants named Syn078, E7k078, Cec078, and Buf078 (Figure 6a). The fusion of peptides significantly enhanced the lytic activity of LysPd078. Specifically, treatment with 6.25 μg/mL Buf078 reduced the number of *S. typhimurium* by ~3 log, whereas LysPd078 at the same concentration achieved a reduction of less than 0.5 log (Figure 6b). Additionally, the time–kill curve showed that Syn078 reduced the number of viable *S. typhimurium* cells to below the detection line within 30 min, significantly exceeding the effect of LysPd078 at the same rate (Figure 6c).

The bactericidal activity of the modified LysPd078 variants was further tested under different conditions, including in serum and salinity. The engineered endolysins exhibited a higher tolerance to NaCl compared to LysPd078 (Figure 7a). The lytic activity of endolysins was also tested in human serum at concentrations ranging from 1% to 50%. While the activity of LysPd078 was significantly inhibited and only minimal activity was retained at low serum concentrations, the engineered endolysins maintained high activity against *S. typhimurium* in serum. Cec078, in particular, showed the same potent activity in 50% serum as in the non-serum control (Figure 7b). Considering that foodborne pathogens can survive for extended periods under cold storage, we tested the bactericidal effect of the engineered endolysins at 4 °C. Although low temperature markedly reduced their bactericidal activity, all engineered endolysins still exhibited significantly higher bactericidal activity compared to LysPd078 (Figure 7c). These results confirmed that the bactericidal activity of LysPd078 was substantially enhanced by peptide fusion.

Next, we evaluated the effect of engineered endolysins on a pre-formed biofilm of *S. typhimurium*, *E. coli*, and *L. monocytogenes*, respectively. CFU plate counts showed the activity of the engineered endolysins against bacteria in the biofilms (Figure 8a–c). The engineered endolysins effectively reduced the number of viable bacteria within the biofilms. At a concentration of 50 μg/mL, the engineered endolysin was sufficient to disrupt the biofilm, while 200 μg/mL caused significant biofilm destruction. Notably, E7k078 at 200 μg/mL eliminated over 99.9% of viable bacteria in biofilms formed by *S. typhimurium*, *E. coli*, and *L. monocytogenes*, respectively. SEM analysis revealed morphological changes in cells before and after treatment with E7k078. Following treatment, most cells showed shrinkage and deformation, with multiple pores on the surface and the leakage of cell contents, leaving only empty shells (Figure 8d–f).

### 2.4. Evaluation of Endolysin Killing Activity Using a Mung Bean Decontamination Model

LysPd078 and engineered endolysins were used to treat contaminated mung beans to assess the potential of endolysins for microbial control during sprout production. Mung beans were artificially contaminated with *S. typhimurium*, *E. coli*, or *L. monocytogenes*, and then treated with endolysins or buffer. After soaking the contaminated mung beans in 50 μg/mL or 100 μg/mL of endolysins, no bacteria were detected in the soak solutions, corresponding to an average reduction of 4–5 log in viable bacterial load compared to the buffer-treated controls (Figure 9a,b). Further sonication of the contaminated mung beans revealed that the bacterial titer in the buffer-treated group was approximately 4 log units/g (Figure 9a). In contrast, after treatment with 100 μg/mL of LysPd078 and engineered endolysins, no viable bacteria were detected in the sonicated mung beans (Figure 9b). Similarly, 50 μg/mL of Syn078, E7k078, and Cec078 exhibited outstanding decontamination capabilities (Figure 9a). This suggested that these endolysin-treated groups successfully eliminated approximately 4–5 log levels of bacteria from the contaminated mung beans. These results demonstrate that LysPd078 and engineered endolysins significantly reduced the bacterial count in contaminated mung beans, highlighting their potential for microbial contamination control during sprout production.

## 3. Discussion

*S. typhimurium*, *E. coli*, and *L. monocytogenes* are foodborne pathogens responsible for an increasing number of foodborne outbreaks, posing a significant threat to human health and causing considerable economic losses [1,47]. Among the various antimicrobial strategies, bacteriophage endolysins have emerged as promising alternatives for the biological control of foodborne pathogens in the food industry [48]. Common foodborne pathogens include various Gram-positive and Gram-negative bacteria, but previous studies have typically focused on just one of them, which is insufficient to effectively control foodborne outbreaks. In this study, we evaluated the potential of novel endolysins and engineered derivatives to control *S. typhimurium*, *E. coli*, and *L. monocytogenes* on mung bean sprouts, aiming to assess their biocontrol potential during sprout production.

LysPd078 and LysPd138 exhibited peptidoglycan hydrolytic activity against *S. typhimurium*, *E. coli*, and *L. monocytogenes* (tested Gram-negative and Gram-positive bacteria), and demonstrated superior bactericidal activity without the need for outer membrane permeabilizers. Although LysPd144 and LysPd157 did not exhibit peptidoglycan hydrolytic activity, they still displayed bactericidal activity, possibly due to the presence of other mechanisms independent of catalysis [49], like Ply17 [50]. Besides their potent activity, these endolysins retained their bactericidal properties across a wide pH range (4 to 10). In addition, the results showed that heat treatment up to 100 °C did not reduce the bactericidal activity of these endolysins, and they maintained effective activity after being stored for 18 months at 4 °C, 22 °C, and 37 °C. This is particularly noteworthy because most endolysins are unstable at temperatures above 50 °C [49,51,52], and reported endolysins typically show only short-term stability, ranging from 1 week to 6 months [49]. These features—a broad pH stability, wide temperature tolerance, and excellent storage properties—highlight the strong stability of these endolysins, supporting their versatility and potential for diverse applications and processing conditions. Notably, LysPd078 exhibited broad-spectrum bactericidal effects against different strains of *S. typhimurium*, *E. coli*, and *Listeria*. These results highlight the commercial potential of LysPd078 in applications such as enzyme engineering, food processing, and biotechnological production.

Despite these advantages, LysPd078 is highly sensitive to NaCl, losing most of its activity in the presence of NaCl. To address this challenge, particularly in complex environments like NaCl and serum, we engineered LysPd078 by fusing it with four antimicrobial peptides (AMPs) selected from the DBAASP database. The resulting engineered endolysins, Syn078, E7k078, Cec078, and Buf078, exhibited significantly enhanced activity, bactericidal kinetics, and salt tolerance. Notably, they completely eradicated viable bacteria in serum. This is a notable achievement given that a major challenge for the application of Gram-negative endolysins is the reduced or abolished killing activity in serum [35,37,53]. Foodborne pathogens, such as *S. typhimurium*, *E. coli*, and *L. monocytogenes*, have been shown to form biofilms [54]. Persistent biofilms often hinder the penetration of antimicrobial agents, limiting their efficacy [55]. Notably, our engineered endolysins were able to penetrate biofilm matrices, overcoming the common challenge of biofilm-associated antimicrobial resistance, and effectively eliminating most viable bacteria within the biofilms.

Regarding the microbial contamination of sprouts, most studies have focused solely on controlling *Salmonella* or *E. coli* on contaminated seeds, and have often showed limited efficacy. For example, the addition of *Salmonella* phage SSP6 to alfalfa seeds contaminated with *S. Oranienburg* resulted in an approximately 1 log reduction in viable *Salmonella* [21]. Similarly, *Salmonella* phage SI1 was applied to alfalfa seeds contaminated with *S. Enteritidis*, leading to a 38.3% reduction in viable *Salmonella* cells [24]. Additionally, the FDA-approved phage cocktail SalmoFresh™ reduced *Salmonella* on mung bean seed surfaces by 2.28 log [25]. Other phages, such as vB_EcoM-Sa45lw, showed a 2-log reduction in *E. coli* on mung bean seeds, while *Escherichia* phage Sa157lw reduced *E. coli* and *Salmonella* by 1.1 log and 1.8 log, respectively [56]. In this study, we simultaneously assessed the effect of endolysins on *S. typhimurium* CMCC (B) 50115, *E. coli* O157:H7, and *L. monocytogenes* WHG50001. Our results indicate that LysPd078 and its engineered lysozymes can significantly reduce the bacterial load on contaminated mung bean seeds by up to 4–5 log (over 99.99%).

In conclusion, the native and engineered endolysins in this study exhibit excellent decontaminating activity against *S. typhimurium*, *E. coli*, and *L. monocytogenes* on mung beans, indicating their promising potential for food safety applications.

## 4. Materials and Methods

### 4.1. Bacterial Strains

The bacterial strains (Table 1) used in this study were grown at 37 °C with aeration (200 rpm). The *Listeria* strains were cultured in brain heart infusion medium (BHI, Qingdao Haibo, Qingdao, China), while *Salmonella* and *Escherichia* strains were cultured in lysogeny broth (LB, Qingdao Haibo, Qingdao, China).

### 4.2. Plasmid Construction

An overview of all the antimicrobial peptides and engineered endolysins produced in this study is provided in Table 2. The endolysins (LysPd078, LysPd138, LysPd144, and LysPd157) used in this study were derived from unpublished sequences developed in our laboratory. The engineered endolysins were constructed by fusing a peptide to the 5′ end of LysPd078 using an AGAGAG linker. To facilitate this fusion, homologous recombination sequences were designed: one in the peptide DNA and another in the middle of the pET-28b(+) vector. The DNA sequences of each antimicrobial peptide or endolysin were amplified using specific primers. The PCR product of each antimicrobial peptide or endolysin was purified, quantified, and assembled using the ClonExpress Ultra One Step Cloning Kit (C116-02, Vazyme, Nanjing, China) [57]. This process successfully fused peptides to LysPd078, resulting in recombinant engineered endolysins.

### 4.3. Expression and Purification of Endolysins

The recombinant proteins were expressed as previously described with slight modifications [62]. Briefly, the selected endolysins were expressed in *E. coli* BL21 cells, induced by 0.4 mmol/L IPTG at 16 °C for 16 h. After induction, the cells were harvested and lysed by sonication on ice. Purification was performed following the instructions of HisTrap FF columns (GE Healthcare, Chicago, IL, USA). The proteins were purified by washing and elution using 20 mM and 250 mM imidazole, respectively. Following dialysis against 10 mM HEPES buffer (pH 7.4), the purified proteins were stored at 4 °C until use. Protein concentration was determined using the Pierce BCA Protein Assay Kit (Thermo Scientific, Waltham, MA, USA).

### 4.4. Halo Assay

To prepare the bacterial overlay agarose, 200 mL of overnight culture of each bacterium was collected by centrifugation (8000 rpm, 5 min) and resuspended in 50 mL of 10 mM HEPES buffer (pH 7.4) containing 0.7% agarose. The mixture was autoclaved, poured onto plates, and stored at 4 °C until use [37]. For the halo assay, 5 μL of each protein (100 μg/mL) was spotted onto the plates, which were then incubated at 37 °C and examined for clearing zones after 1 h.

### 4.5. Bactericidal Activity Assays

The bactericidal activity was determined by log-killing assay as described previously [63]. Briefly, an overnight culture of the test bacteria was diluted 1:100 and grown to an optical density of 0.5 at 600 nm (OD_600_), and measured using a microplate reader (Synergy H1, BioTek Instruments, Winooski, VT, USA). The bacterial cells were then harvested, washed, and resuspended in 10 mM HEPES buffer (pH 7.4) to a final concentration of approximately 10⁶ cells/mL. In a 96-well plate, each endolysin was diluted in 50 μL of 10 mM HEPES buffer to the desired final concentration, after which 50 μL of the prepared bacterial suspension was added to each well. The plate was incubated for 1 h at 37 °C. Following incubation, the contents of each well were serially diluted in a 10-fold manner and plated on LB or BHI agar to assess the bacterial viability. The dilution process was carried out immediately to rapidly reduce the endolysin concentrations, preventing potential bacterial killing during dilution and plating. Negative controls were conducted by treating wells with an equivalent volume of HEPES buffer in place of endolysins. In the following experiments, the bactericidal activity was determined according to the above conditions unless otherwise specified.

To evaluate the bactericidal activity of different endolysins against *S. typhimurium* CMCC (B) 50115, *E. coli* O157:H7, and *L. monocytogenes* WHG50001, the bacterial suspensions were treated with 100 μg/mL of each endolysin for 1 h at 37 °C. For the time–kill curves, *S. typhimurium* CMCC (B) 50115 cells were mixed with 100 μg/mL endolysins for varying times (0, 5, 15, 30, and 60 min). A dose-dependent killing assay was performed by treating *S. typhimurium* CMCC (B) 50115 cells with increasing endolysin concentrations (0, 6.25, 12.5, 25, 50, and 100 μg/mL) for 1 h at 37 °C. To test the bactericidal effects of endolysins on different growth phases of bacteria, the bacteria was transferred at a 1:100 ratio into fresh culture medium. After culturing for 2–4 h, bacteria in the log phase (OD_600_ = 0.4–0.6) were obtained. Culturing for 10–12 h yielded bacteria in the stationary phase (OD_600_ = 1.0–1.4) [64,65]. The log and stationary phase cultures of *S. typhimurium* CMCC (B) 50115 were treated with 100 μg/mL endolysins for 1 h at 37 °C.

To evaluate the influence of external factors on enzyme activity, we tested the activity of the enzyme under different pH values, temperatures, in serum, and in salinity. The influence of NaCl on the bactericidal activity was assessed by treating *S. typhimurium* CMCC (B) 50115 cells suspended in HEPES buffer containing various concentrations of NaCl (20, 50, 100, 300, and 1000 mM) with 100 μg/mL endolysins for 1 h at 37 °C. The effect of pH on the bactericidal activity was determined by suspending *S. typhimurium* CMCC (B) 50115 cells in 100 mM sodium acetate buffer with pH 5.0–6.0, 20 mM HEPES buffer with pH 7.0–8.0, or 100 mM Glycine–NaOH buffer at pH 9.0–10.0 and treating them with 100 μg/mL endolysins for 1 h at 37 °C. To evaluate the effect of serum on the bactericidal activity of endolysins, *S. typhimurium* CMCC (B) 50115 cells suspended in HEPES buffer containing various concentrations of serum (2, 10, 20, 50, and 100%) were treated with 100 μg/mL endolysins for 1 h at 37 °C.

Next, we tested the enzyme stability. The bactericidal activity of endolysins at low temperatures was tested by treating *S. typhimurium* cells in HEPES buffer with 100 μg/mL endolysins at 4 °C for 1, 2, 4, and 6 h. To evaluate the thermostability, endolysins were stored at different temperatures (20 °C, 40 °C, 60 °C, 80 °C, and 100 °C) for 1 h, and residual bactericidal activity was determined by exposing *S. typhimurium* CMCC (B) 50115 cells to 100 μg/mL endolysins for 1 h at 37 °C. The storage stability was assessed by storing endolysins at 4 °C, 22 °C, or 37 °C for up to 18 months, followed by measuring residual bactericidal activity by exposing *S. typhimurium* CMCC (B) 50115 cells to 100 μg/mL endolysins for 1 h at 37 °C. The bactericidal activity was presented as the relative inactivation as follows: Relative bactericidal activity (%) = 100 − (Ni/Nn) × 100, where Nn = number of residual cells (in the negative control) and Ni = the number of viable cells (under different conditions).

To further estimate the bactericidal spectrum of LysPd078, different bacterial strains were used to further estimate the lytic spectrum of LysPd078.

### 4.6. Biofilm CFU Assays

The antibiofilm activity of endolysins was evaluated by assessing the reduction in biofilm-associated bacterial numbers in the 96-well plate. The 96-well plates used for biofilm formation were made of polystyrene, a material commonly used for bacterial adhesion and biofilm studies, with a clear, flat bottom, and 300 μL/well [66,67]. For biofilm formation, overnight cultures of *S. typhimurium* CMCC (B) 50115, *E. coli* O157:H7, and *L. monocytogenes* WHG50001 were diluted 1:1000 in LB or BHI broth supplemented with 1% glucose (*w*/*v*) to promote biofilm development [68,69]. To minimize evaporation and maintain consistent conditions, the outer wells of the plate were filled with sterile phosphate-buffered saline (PBS) or water. The plates were then covered with a breathable sealing film and incubated at 37 °C for 24 h with continuous shaking at 100 rpm to facilitate uniform bacterial growth and biofilm formation. The resultant biofilms were washed once, then treated with either HEPES buffer or endolysins at concentrations of 50 μg/mL and 200 μg/mL at 37 °C for 2 h. Viable bacterial cell counts within the biofilms were determined by re-suspending adherent bacterial cells through vigorous pipetting and vortexing, followed by serial dilution and plating. The antibiofilm activity of the endolysins was quantified as the relative inactivation using the following formula: Clearance % = 100 − (Ni/Nn) × 100, where Nn = number of residual cells (in the negative control) and Ni = the number of viable cells (under different conditions).

### 4.7. Scanning Electron Microscopy (SEM)

Overnight cultures of *Salmonella typhimurium* CMCC (B) 50115, *Escherichia coli* O157:H7, and *Listeria monocytogenes* WHG50001 were diluted 1:1000 in LB or BHI broth containing 1% glucose (*w*/*v*) and inoculated into a 24-well plate, with a glass coverslip in each well. The plate was incubated at 37 °C for 24 h with continuous shaking at 100 rpm to promote uniform bacterial growth and biofilm formation. The resulting biofilms were washed once, then treated with HEPES buffer or E7k078 at a concentration of 200 μg/mL at 37 °C for 2 h. The biofilms were then washed twice with HEPES buffer and fixed in 2.5% glutaraldehyde overnight at 4 °C [70]. Subsequently, the coverslips were dehydrated through a series of increasing ethanol concentrations (30%, 50%, 60%, 70%, 80%, 90%, and 100%). Finally, the samples were gold-coated (Hummer VI; Technic Inc., Anaheim, CA, USA) and observed under SEM (SU8010, Hitachi Ltd., Chiyoda, Tokyo, Japan) at an accelerating voltage of 3 kV and a magnification of ×5000.

### 4.8. Mung Bean Contamination Model

Mung bean seeds were purchased from a local store and disinfected by soaking in 75% ethanol for 1 min, followed by rinsing three times with sterile water and air drying at room temperature in a biosafety cabinet, as previously described [22,25]. *S. typhimurium* CMCC (B) 50115, *E. coli* O157:H7, and *L. monocytogenes* WHG50001 were grown to the mid-exponential phase before seed inoculation. The seeds were immersed in a bacterial suspension (~10^7^ cells/mL) for 1 h at room temperature and then air dried in a biosafety cabinet. The contaminated seeds (~0.5 g) were subsequently soaked in either HEPES buffer or endolysins at concentrations of 50 μg/mL or 100 μg/mL for 1 h at 37 °C. After treatment, each seed was transferred to a 1.5 mL tube containing 500 μL of PBS and subjected to ultrasonication for 1 min (ultrasonic bath, 40 KHz, 22 °C, SB-5200DTD, SCIENTZ, Ningbo, China). Both the soak solutions and the ultrasonicated samples were serially diluted and plated on LB or BHI agar to enumerate the bacteria in bacterial CFUs.

## Figures and Tables

**Figure 1 ijms-26-02047-f001:**
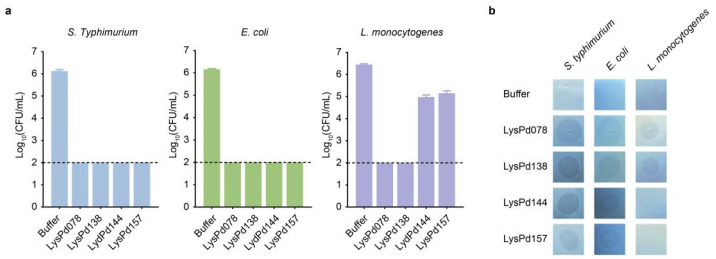
Activity of endolysins. (**a**) Antibacterial activity of each endolysin (100 μg/mL) against *S. typhimurium* CMCC (B) 50115, *E. coli* O157:H7, and *L. monocytogenes* WHG50001. (**b**) Peptidoglycan hydrolytic capacity of each endolysin (100 μg/mL) against *S. typhimurium* CMCC (B) 50115, *E. coli* O157:H7, and *L. monocytogenes* WHG50001. The dashed line indicates the detection limit. Data are presented as mean ± S.D.

**Figure 2 ijms-26-02047-f002:**
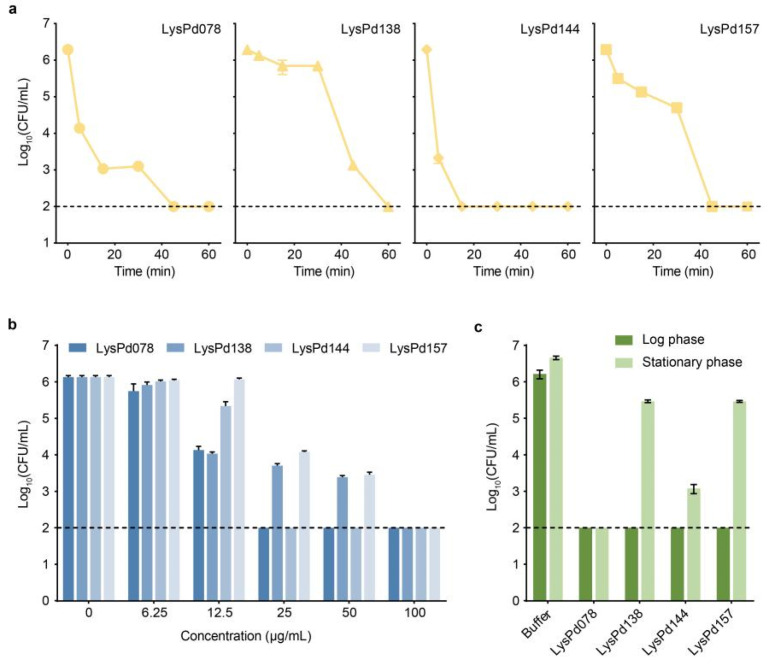
Bactericidal activity of endolysins against *S. typhimurium* CMCC (B) 50115 under different conditions. (**a**) Time–kill curve of each endolysin (100 μg/mL). (**b**) Dose-dependent bactericidal activity of each endolysin at varying concentrations. (**c**) Antibacterial activities of each endolysin (100 μg/mL) against the *S. typhimurium* cells in different growth phases. The dashed line indicates the detection limit. Data are expressed as mean ± S.D.

**Figure 3 ijms-26-02047-f003:**
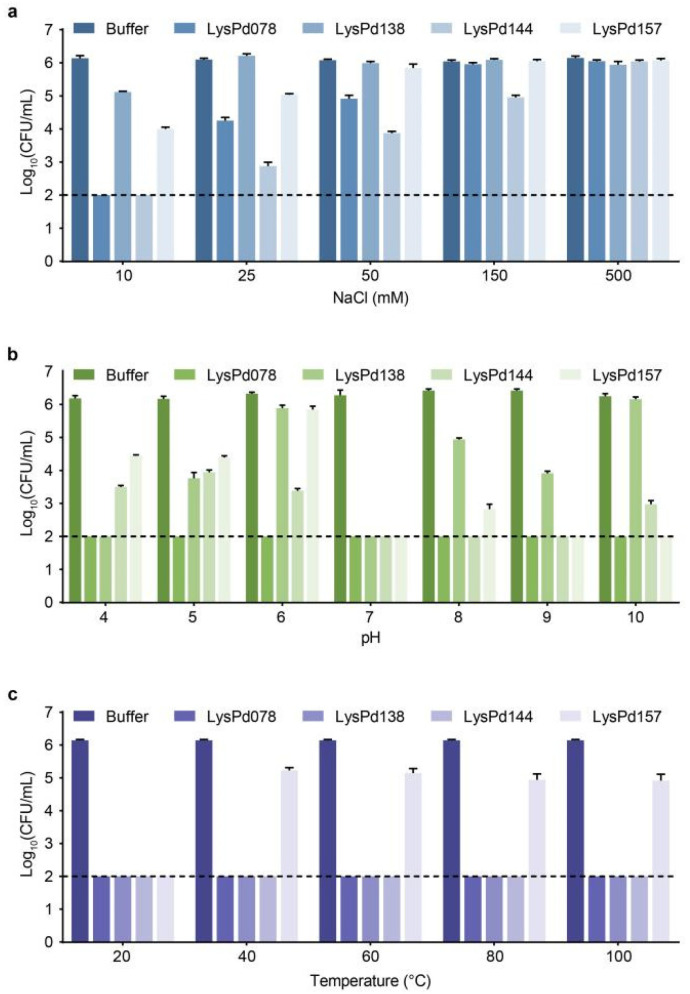
Biochemical characterization of endolysins. The effect of NaCl (**a**), pH (**b**), and temperature (**c**) on the bactericidal activity of each endolysin (100 μg/mL) against *S. typhimurium* CMCC (B) 50115. The dashed line indicates the detection limit. Data are expressed as mean ± S.D.

**Figure 4 ijms-26-02047-f004:**
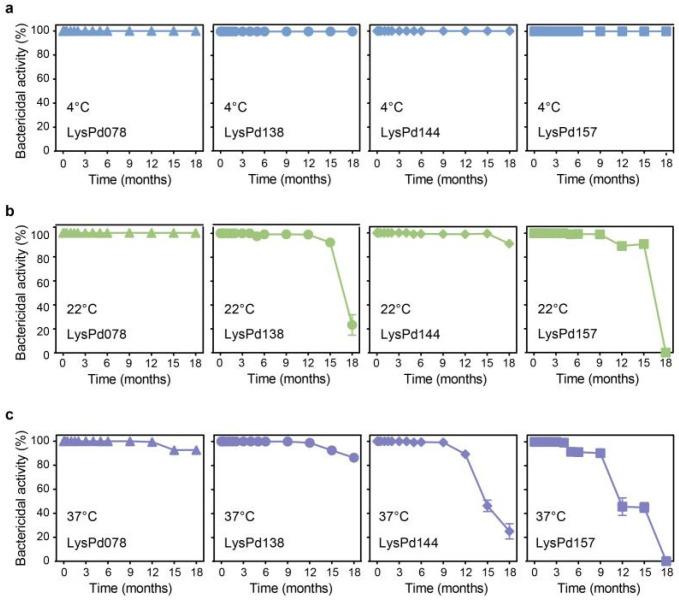
Storage stability of each endolysin (100 μg/mL) at 4 °C (**a**), 22 °C (**b**), and 37 °C (**c**). Data are expressed as mean ± S.D.

**Figure 5 ijms-26-02047-f005:**
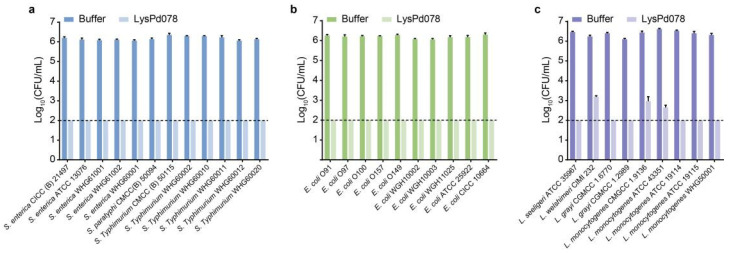
The bactericidal activity spectrum of LysPd078 (100 μg/mL) against *Salmonella* (**a**), *E. coli* (**b**), and *Listeria* (**c**) species. The dashed line indicates the detection limit. Data are expressed as mean ± S.D.

**Figure 6 ijms-26-02047-f006:**
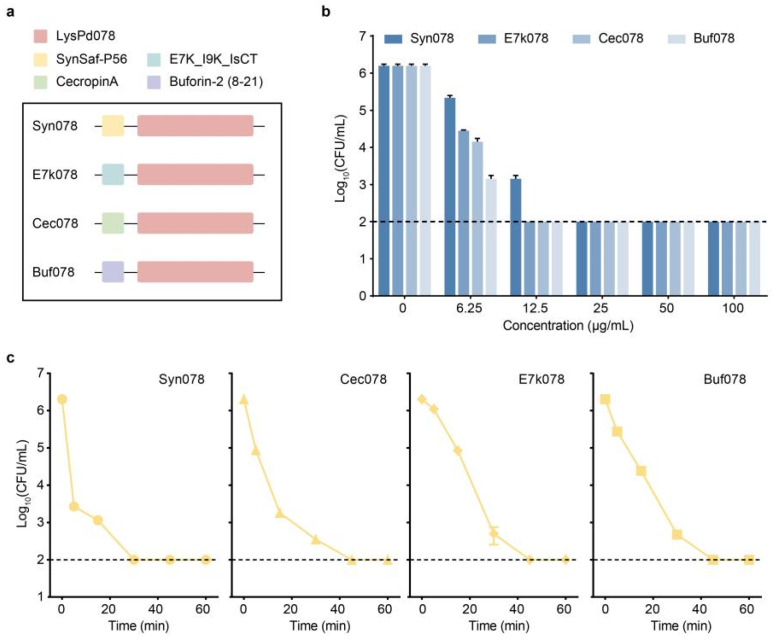
The bactericidal activity of the engineered endolysins against *S. typhimurium* CMCC (B) 50115. (**a**) Schematic representation of the engineered endolysins. LysPd078 was selected for modification with different peptides. (**b**) Bactericidal activity of each artiendolysin under different concentrations. (**c**) Time–kill curve of each artiendolysin (100 μg/mL). The dashed line indicates the detection limit. Data are expressed as mean ± S.D.

**Figure 7 ijms-26-02047-f007:**
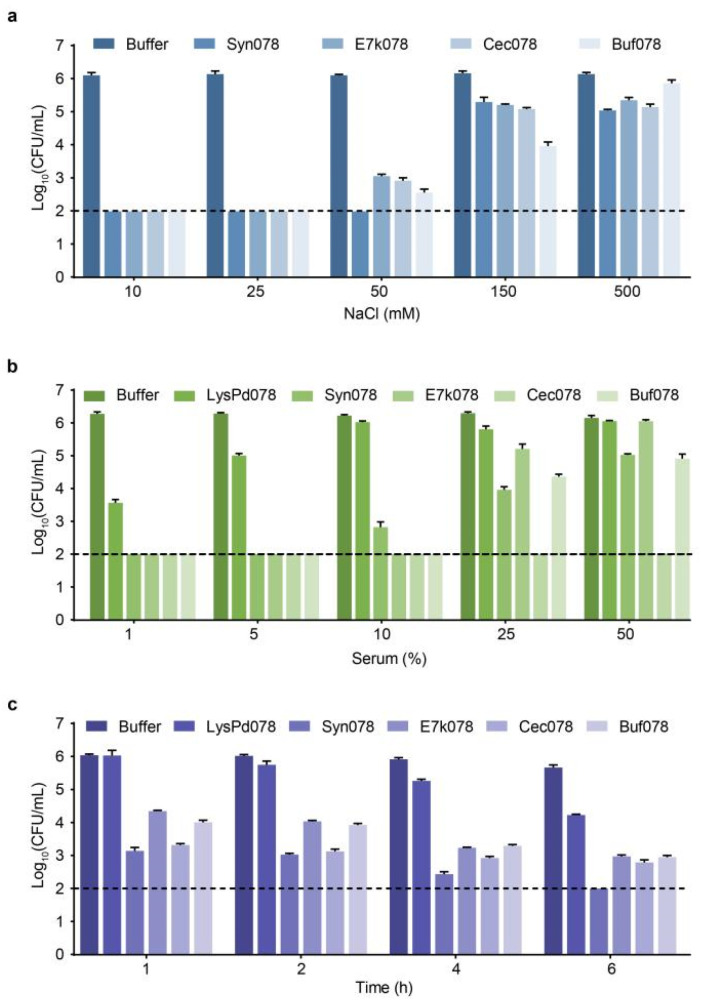
Biochemical characterization of engineered endolysins. The effect of NaCl (**a**), serum (**b**), and 4 °C (**c**) on the bactericidal activity of each engineered endolysin (100 μg/mL) against *S. typhimurium* CMCC (B) 50115. The dashed line indicates the detection limit. Data are expressed as mean ± S.D.

**Figure 8 ijms-26-02047-f008:**
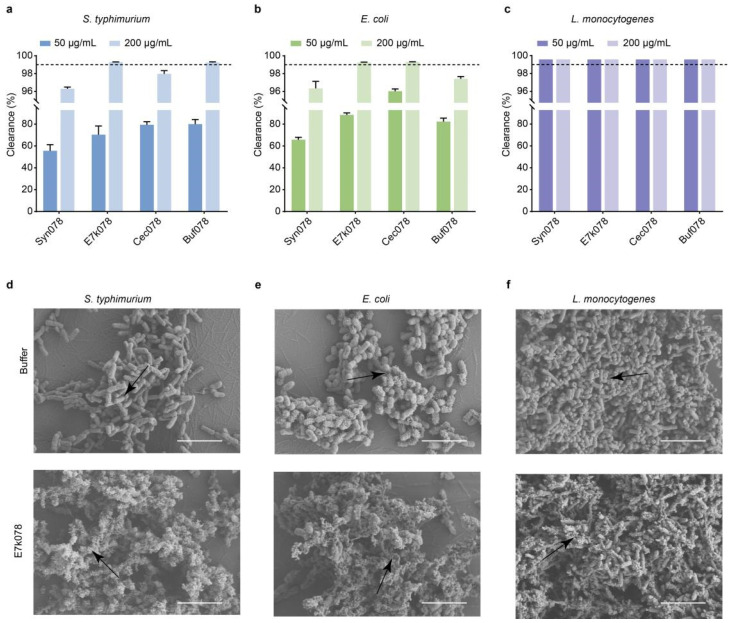
Antibiofilm activity of each endolysin against *S. typhimurium* CMCC (B) 50115 (**a**), *E. coli* O157:H7 (**b**), and *L. monocytogenes* WHG50001 (**c**). The dashed line indicates 99.9% clearance. Data are expressed as mean ± S.D. SEM images of the morphological changes in biofilms against *S. typhimurium* CMCC (B) 50115 (**d**), *E. coli* O157:H7 (**e**), and *L. monocytogenes* WHG50001 (**f**) treated with E7k078 (200 μg/mL). Scale bar, 5 μm.

**Figure 9 ijms-26-02047-f009:**
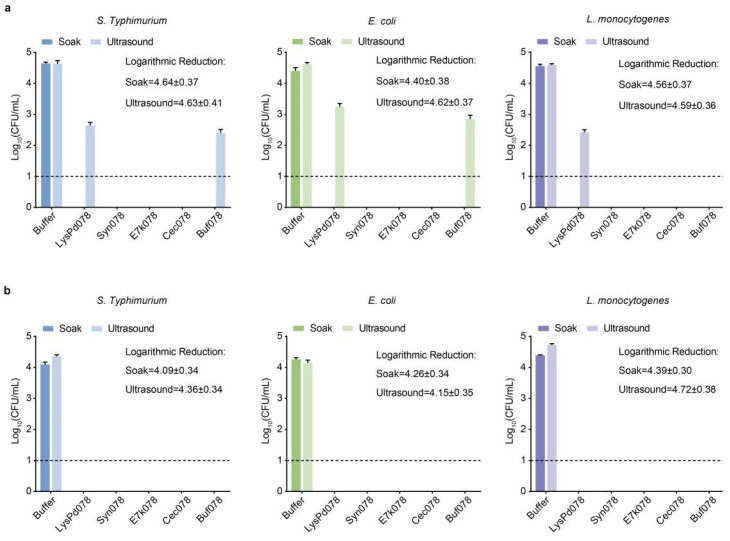
Endolysin decontamination of contaminated mung beans. Reduction in bacterial loads on contaminated mung beans after each treatment with 50 μg/mL (**a**) and 100 μg/mL (**b**). Logarithmic reduction = CFU (Average ± Standard Error). The dashed line indicates the detection limit. Data are expressed as mean ± S.D.

**Table 1 ijms-26-02047-t001:** Bacterial strains used in this study.

Bacterial Species	Strain	Source ^a^
Gram-positive bacteria		
*Listeria grayi*	CGMCC 1.2989	2
*Listeria grayi*	CGMCC 1.6770	2
*Listeria welshimeri*	CIMI 232	1
*Listeria seeligeri*	ATCC 35967	1
*Listeria monocytogenes*	WHG50001	1
*Listeria monocytogenes*	ATCC 19114	1
*Listeria monocytogenes*	ATCC 19115	1
*Listeria monocytogenes*	CMGCC 1.9136	2
*Listeria monocytogenes*	ATCC 43351	1
Gram-negative bacteria		
*Salmonella paratyphi*	CMCC 50094	3
*Salmonella enterica*	CICC 21497	3
*Salmonella enterica*	ATCC 13076	1
*Salmonella enterica*	WHG61001	1
*Salmonella enterica*	WHG61002	1
*Salmonella enterica*	WHG60001	1
*Salmonella typhimurium*	CMCC 50115	3
*Salmonella typhimurium*	WHG60002	1
*Salmonella typhimurium*	WHG60010	1
*Salmonella typhimurium*	WHG60011	1
*Salmonella typhimurium*	WHG60012	1
*Salmonella typhimurium*	WHG60020	1
*Escherichia coli*	O91	1
*Escherichia coli*	O97	1
*Escherichia coli*	O100	1
*Escherichia coli*	O157:H7	1
*Escherichia coli*	O149	1
*Escherichia coli*	WGH10002	1
*Escherichia coli*	WGH10003	1
*Escherichia coli*	WGH11025	1
*Escherichia coli*	ATCC 25922	1
*Escherichia coli*	CICC 10664	3
*Escherichia coli*	BL21 (DE3)	1

^a^ Source: 1. Laboratory collection, Key Laboratory of Special Pathogens and Biosafety, Wuhan Institute of Virology, China. 2. Purchased from China General Microbiological Culture Collection Center, China. 3. Purchased from Guangdong Culture Collection Center, China.

**Table 2 ijms-26-02047-t002:** Overview of engineered endolysins.

Name	Endolysin	linker	Peptide
Syn078	LysPd078	AGAGAG	SynSaf-P56 [58]
E7k078	LysPd078	AGAGAG	E7K_I9K_IsCT [59]
Cec078	LysPd078	AGAGAG	CecropinA [60]
Buf078	LysPd078	AGAGAG	Buforin-2 (8-21) [61]

## Data Availability

The authors confirm that all data underlying the findings are fully available without restriction. All relevant data are within the paper.

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
