# Peer review of "Endolysins as Effective Agents for Decontaminating S. typhimurium, E. coli, and L. monocytogenes on Mung Bean Seeds"

_ijms, 2025, doi:10.3390/ijms26052047_

Round 1
Reviewer 1 Report
Comments and Suggestions for Authors
First of all, I would like to begin with an appreciation of the authors' extensive work.
In the submitted manuscript, Lysins as Effective Agents for Decontaminating S. Typhimurium, E. coli, and L. monocytogenes on Mung Bean Seeds, authored by Fangfang Yao, Jiajun He, Raphael Nyaruaba, Hongping Wei, and Yuhong Li, the antibacterial potential of novel lysins and their engineered derivatives was evaluated for the control of S.Typhimurium, E. coli, and L. monocytogenes on mung bean sprouts.
In my opinion, the obtained results are original and bring novel and relevant insights to the scientific community in this field and beyond. The manuscript aligns with the scope of the IJMS journal and can be published after some minor revisions and clarifications.
A few questions and suggestions:
Lines 15-16 Abstract
Please write the full name of the microorganisms used in the study when they first appear in the text—here, in the abstract section.
When I first read the text, I initially thought it was referring to the amino acid lysine. To avoid confusion and for clarity, please specify that this refers to endolysins used in antibacterial therapy, or rephrase it in a way that makes this clear. For me, 'bactericidal' means complete bacterial eradication. What does 'potent bactericidal activity' mean to you? I believe certain passages in the abstract should be supported with clear, quantitative data to be more convincing. For example, by how much was the bacterial load reduced, and so on.
Lines 91-98 Introduction
I believe this passage needs improvement, as it presents conclusions, making it unsuitable for this section. Please focus only on the aim of the study.
Line 105 Results
What is the detection limit? I think it should also be written in the text.
Line 120. Characterization of lysins
For better clarity, I suggest presenting the dose-response experiment (Fig. 2b) first, followed by the time-kill curves (Fig. 2a) and the comparison between log-phase and stationary-phase cells (Fig. 2c).
Line 157
I do not agree with the use of the term isolates in this study, since the strains are precisely known.
Line 315
E. coli – please, use italic font
Line 330 Bactericidal activity assays
This methodology completely lost me. The organization is unclear, the incubation procedure is repeatedly mentioned, and some tests (pH, temperature, etc.) could perhaps be summarized in a table? If you allow me to make another suggestion, please organize the information more clearly by grouping the tests into different categories: antibacterial activity, influence of external factors (pH, temperature, salinity, etc.), stability tests.
Line 333
remove the space
Lines 343-344
Please connect the two sentences for coherence and clarity.
Line 346
Please check throughout the text …..S. Typhimurium 345 CMCC (B) 50,115 ?
Line 397 place the period at the end of the sentence
Please carefully check the references and ensure they comply with the journal's requirements. You have multiple variations in formatting:
Fong, K. et al. Characterization of Four Novel Bacteriophages Isolated from British Columbia for Control of Non-typhoidal Sal- 462 monella in Vitro and on Sprouting Alfalfa Seeds. Front Microbiol 8, 2193, doi:10.3389/fmicb.2017.02193 (2017)
Liao, Y.T., Zhang, Y., Salvador, A., Harden, L.A. & Wu, V.C.H. Characterization of a T4-like Bacteriophage vB_EcoM-Sa45lw as 457 a Potential Biocontrol Agent for Shiga Toxin-Producing Escherichia coli O45 Contaminated on Mung Bean Seeds. Microbiol Spectr 458 10, e0222021, doi:10.1128/spectrum.02220-21 (2022).
Feng, G., Churey, J.J. & Worobo, R.W. Thermal inactivation of Salmonella and Escherichia coli O157:H7 on alfalfa seeds. J Food 460 Prot 70, 1698-1703, doi:10.4315/0362-028x-70.7.1698 (2007).
Author Response
In my opinion, the obtained results are original and bring novel and relevant insights to the scientific community in this field and beyond. The manuscript aligns with the scope of the IJMS journal and can be published after some minor revisions and clarifications.
Response: We sincerely appreciate the reviewer's kind and constructive comments, as well as the time and effort dedicated to improving our manuscript. We have carefully addressed each of the reviewer's suggestions point by point in our response. We hope that the revised manuscript now meets the publication standards of the journal. Thank you once again for your valuable feedback.
Comment 1: Lines 15-16 Abstract
Please write the full name of the microorganisms used in the study when they first appear in the text—here, in the abstract section. When I first read the text, I initially thought it was referring to the amino acid lysine. To avoid confusion and for clarity, please specify that this refers to endolysins used in antibacterial therapy, or rephrase it in a way that makes this clear. For me, 'bactericidal' means complete bacterial eradication. What does 'potent bactericidal activity' mean to you? I believe certain passages in the abstract should be supported with clear, quantitative data to be more convincing. For example, by how much was the bacterial load reduced, and so on.
Response 1: We thank the reviewer for this suggestion. In the abstract section, we have revised the microorganisms used in the study by their full names (Page 1, Line 16-17). To avoid confusion and ensure clarity, we have revised 'lysin' throughout the article to 'endolysin', which is also a widely used term for bacteriophage lysins (https://doi.org/10.1111/1541-4337.13145; https://doi.org/10.3390/antibiotics9020074; https://doi.org/10.1016/j.ijpharm.2020.119833). In this paper, 'bactericidal' means a reduction in bacterial count, and 'potent bactericidal activity' refers to a significant reduction in bacterial numbers. We fully agree with the reviewer's suggestion and have rewritten the abstract, adding clear quantitative data to enhance its persuasiveness (Page 1, Line 16-31).
Comment 2: Lines 91-98 Introduction
I believe this passage needs improvement, as it presents conclusions, making it unsuitable for this section. Please focus only on the aim of the study.
Response 2: We thank the reviewer for this suggestion. We have made the corresponding revisions in the revised manuscript (Page 3, Line 102-110).
Comment 3: Line 105 Results
What is the detection limit? I think it should also be written in the text.
Response 3: We thank the reviewer for this suggestion. We performed bacterial CFU counting, and 10 μL of the dilution were spotted at each dilution level. Therefore, even if no bacterial growth was observed on the plate, we could only conclude that the bacterial concentration is below 100 CFU/mL. Here, the detection limit refers to the minimum level at which bacterial presence can be detected. we have made the corresponding revisions in the revised manuscript (Page 4, Line 117).
Comment 4: Line 120. Characterization of lysins
For better clarity, I suggest presenting the dose-response experiment (Fig. 2b) first, followed by the time-kill curves (Fig. 2a) and the comparison between log-phase and stationary-phase cells (Fig. 2c).
Response 4: We thank the reviewer for this suggestion. For food applications, where pre-processing time is often constrained, rapid bactericidal kinetics is a critical parameter of interest. Therefore, we want to prioritize presenting the time-kill curve results to evaluate the efficacy of bacterial inactivation over time.
Comment 5: Line 157
I do not agree with the use of the term isolates in this study, since the strains are precisely known.
Response 5: We thank the reviewer for this suggestion. These points have been revised as suggested in the revised manuscript (Page 6, Line 173; Page 7, Line 175).
Comment 6: Line 315
E. coli – please, use italic font
Response 6: Thank the reviewer for pointing this error. This point has been revised as suggested in the revised manuscript (Page 12, Line 344).
Comment 7: Line 330 Bactericidal activity assays
This methodology completely lost me. The organization is unclear, the incubation procedure is repeatedly mentioned, and some tests (pH, temperature, etc.) could perhaps be summarized in a table? If you allow me to make another suggestion, please organize the information more clearly by grouping the tests into different categories: antibacterial activity, influence of external factors (pH, temperature, salinity, etc.), stability tests.
Response 7: We thank the reviewer for this suggestion. We added further details in the Methods section (Page 13-14, Line 372-412).
Comment 8: Line 333
remove the space
Response 8: Thank the reviewer for pointing this error. This point has been revised as suggested in the revised manuscript (Page 13, Line 362).
Comment 9: Lines 343-344
Please connect the two sentences for coherence and clarity.
Response 9: We thank the reviewer for this suggestion. This point has been revised as suggested in the revised manuscript (Page 13, Line 372-374).
Comment 10: Line 346
Please check throughout the text …..S. Typhimurium 345 CMCC (B) 50,115 ?
Response 10: Thank the reviewer for pointing this error. These points have been revised as suggested in the revised manuscript.
Comment 11: Line 397 place the period at the end of the sentence
Response 11: Thank the reviewer for pointing this error. This point has been revised as suggested in the revised manuscript (Page 14, Line 458).
Comment 12: Please carefully check the references and ensure they comply with the journal's requirements. You have multiple variations in formatting:
Fong, K. et al. Characterization of Four Novel Bacteriophages Isolated from British Columbia for Control of Non-typhoidal Sal- 462 monella in Vitro and on Sprouting Alfalfa Seeds. Front Microbiol 8, 2193, doi:10.3389/fmicb.2017.02193 (2017)
Liao, Y.T., Zhang, Y., Salvador, A., Harden, L.A. & Wu, V.C.H. Characterization of a T4-like Bacteriophage vB_EcoM-Sa45lw as 457 a Potential Biocontrol Agent for Shiga Toxin-Producing Escherichia coli O45 Contaminated on Mung Bean Seeds. Microbiol Spectr 458 10, e0222021, doi:10.1128/spectrum.02220-21 (2022).
Feng, G., Churey, J.J. & Worobo, R.W. Thermal inactivation of Salmonella and Escherichia coli O157:H7 on alfalfa seeds. J Food 460 Prot 70, 1698-1703, doi:10.4315/0362-028x-70.7.1698 (2007).
Response 12: We thank the reviewer for this suggestion. This point has been revised as suggested in the revised manuscript.
Reviewer 2 Report
Comments and Suggestions for Authors
The work is novel, interesting, and advances the field of microbiology and safety. The experiments are well conducted, and the manuscript reads well. Please refer to some comments and suggestions below:
1. Please elaborate on how bacterial cells in the growth and stationary phases are obtained and the experimental technique used to determine cell state.
2. The method for creating biofilms in 96-well plates should be described in more detail, including the material of the 96-well plates. Additionally, images of biofilm formation and disruption would enhance Figure 8.
3. An explanation is needed for why mung beans require sonication. Furthermore, details on the sonication operating conditions should be provided, including temperature control (e.g., ice bath), frequency, voltage, and the model of the equipment used.
Overall, an excellent scientific contribution.
Author Response
Comment: The work is novel, interesting, and advances the field of microbiology and safety. The experiments are well conducted, and the manuscript reads well. Please refer to some comments and suggestions below:
Response: We sincerely thank the reviewer for the positive feedback and for recognizing the novelty and significance of our work in advancing the field of microbiology and safety. We are also grateful for the constructive comments and suggestions provided below, which have helped us further improve the manuscript. We have carefully addressed each point, and our responses are detailed below. Thank you once again for your valuable input.
Comment 1: Please elaborate on how bacterial cells in the growth and stationary phases are obtained and the experimental technique used to determine cell state.
Response 1: We thank the reviewer for the kind comments. To obtain bacterial cultures in the growth and stationary phases, we initially inoculated the bacteria into fresh liquid growth medium and incubated them overnight at 37°C with shaking at 220 rpm. Subsequently, the bacteria were subcultured at a 1:100 ratio into fresh medium. After 2-4 hours of incubation, the optical density at 600 nm (OD600) reached 0.4-0.6, indicating the exponential growth phase (https://doi.org/10.1038/srep38828; https://doi.org/10.1128/aem.00446-16). Following 10-12 hours of incubation, the OD600 values ranged between 1.0 and 1.4, signifying the stationary phase (https://doi.org/10.1038/srep38828). Throughout this process, the bacterial growth status was monitored using a Synergy H1 microplate reader to measure the OD600. The description was also added in the revised manuscript (Page 13, Line 381-386).
Comment 2: The method for creating biofilms in 96-well plates should be described in more detail, including the material of the 96-well plates. Additionally, images of biofilm formation and disruption would enhance Figure 8.
Response 2: We thank the reviewer for the kind comments. The 96-well plates used for biofilm formation were made of polystyrene, clear, flat bottom, 300 μL/well, a material commonly used for bacterial adhesion and biofilm studies (https://doi.org/10.1155/2014/231083; https://doi.org/10.1007/978-1-0716-0459-5_19). For biofilm formation, overnight cultures of S. Typhimurium CMCC (B) 50115, E. coli O157:H7 , and L. monocytogenes WHG50001 were diluted 1:1000 in LB or BHI broth supplemented with 1% glucose (w/v) to promote biofilm development. The diluted bacterial suspensions (100 μL/well) were inoculated into the 96-well plates. To minimize evaporation and maintain consistent conditions, the outer wells of the plate were filled with sterile phosphate-buffered saline (PBS) or water. The plates were then covered with a breathable sealing film and incubated at 37°C for 24 hours with continuous shaking at 100 rpm to facilitate uniform bacterial growth and biofilm formation. The description is also added in the revised manuscript (Page 14, Line 410-420). We fully agree that images of biofilm formation and disruption would enhance Figure 8. Additional experiments were conducted to evaluate the effects of bacteria after endolysin treatment using scanning electron microscopy (SEM). The results and description were added in the revised manuscript (Figure 8 d-f; Page 8, Line 220-225; Page, Line 229-231; Page 14, Line 433-444).
Comment 3: An explanation is needed for why mung beans require sonication. Furthermore, details on the sonication operating conditions should be provided, including temperature control (e.g., ice bath), frequency, voltage, and the model of the equipment used.
Response 3: We thank the reviewer for the kind comments. When mung beans are soaked in water containing bacteria, the bacteria adhere to the surface of the beans. After lysozyme treatment, the bacteria may be killed or inhibited, but the dead or damaged bacteria still remain attached to the surface of the mung beans. Ultrasonic treatment (ultrasonic bath, 40KHz, 22°C, SB-5200DTD, SCIENTZ, China) generates tiny bubbles, and the energy released when these bubbles collapse can disrupt the binding force between the bacteria and the surface of the mung beans. This method effectively releases the bacteria (both dead and alive) adhering to the surface of the mung beans into the solution, thereby ensuring the accuracy of subsequent bacterial counts (https://doi.org/10.1016/j.ijfoodmicro.2019.108250). This is a crucial step in evaluating the bactericidal effect of the drug, as it avoids counting errors caused by bacterial adhesion. We added further details in the Methods section (Page 15, Line 455-456).